# The Strange Case of Orotic Acid: The Different Expression of Pyrimidines Biosynthesis in Healthy Males and Females

**DOI:** 10.3390/jpm13101443

**Published:** 2023-09-28

**Authors:** Francesco Chiara, Sarah Allegra, Jacopo Mula, Maria Paola Puccinelli, Giuliana Abbadessa, Giulio Mengozzi, Silvia De Francia

**Affiliations:** 1Laboratory of Clinical Pharmacology “Franco Ghezzo”, Department of Clinical and Biological Sciences, University of Turin, S. Luigi Gonzaga Hospital, 10043 Turin, Italy; 336124@edu.unito.it (F.C.); giuliana.abbadessa@unito.it (G.A.); silvia.defrancia@unito.it (S.D.F.); 2Department of Medical Sciences, University of Turin, Amedeo di Savoia Hospital, 10149 Turin, Italy; jacopo.mula@unito.it; 3Laboratory of Clinical Biochemistry “Baldi e Riberi”, Metabolic Diseases Unit, AOU Città della Salute e della Scienza di Torino (TO), 10126 Turin, Italy; mpuccinelli@cittadellasalute.to.it (M.P.P.); gmengozzi@cittadellasalute.to.it (G.M.)

**Keywords:** orotic acid, pyrimidine biosynthesis, LC-MS/MS, sex differences

## Abstract

Orotic acid (OA) is an intermediate metabolite of pyrimidine nucleotide biosynthesis and represents a minor diet constituent. The measurement of urinary orotic acid is useful in confirming the diagnosis of hereditary metabolic diseases. Moreover, it could be of interest to know how the physiological concentration of this metabolite changes in relation to different conditions of clinical normality. The purpose of this study was to determine the orotic acid concentration in the urine of healthy patients, to observe normal oroticuria and to evaluate if the expression of pyrimidine intermediate biosynthesis differs between healthy males and females. The orotic acid concentration in urine was performed via the ICH M10-validated analytical method. Unexpectedly, females showed a greater oroticuria than males in pediatric age (0–10); conversely, we did not find significant differences until 70 years of age. The LC-MS/MS method was suitable for use in the differential diagnosis of hereditary metabolic disease and metabolic monitoring of anticancer drug-induced toxicity. The analytical protocol was found to be rapid and ideal, and was used in the routine analysis of a clinical chemistry laboratory. The biochemical aspects related to the expression of pyrimidine biosynthesis should be further investigated in light of the obtained results.

## 1. Introduction

Orotic acid (OA) (2,4-dioxo-1H-pyrimidine-6-carboxylic acid; vitamin B13) is an intermediate metabolite of pyrimidine nucleotide biosynthesis and represents a minor diet constituent. The precursors of OA in human metabolism are the cytosolic CP and CA via dihydroorotate, a biosynthesis catalyzed by a CAD gene encoding multifunctional enzyme [1,2]. The multimeric protein called UMP synthase is constituted by two domains that catalyze UMP synthesis: OPRTase (EC 2.4.2.10) and OMPdecase (EC 4.1.1.23) [3]. The complete pathway of OA biosynthesis is reported in Figure 1. The step (5), represented in Figure 1, is directly involved in the metabolism of 5-FU because this anticancer drug is a competitive substrate of OPRTase [4]. In particular, the transferase activity of OPRTase multicomplex enzyme is inhibited by 5-FU at 59% level of control [5]. On the other hand, OPRTase is involved in a variety of metabolic disorders, such as congenital orotic aciduria. Consequently, the urinary OA is quantified in a clinical routine analysis for a differential diagnosis of hereditary metabolic diseases [6,7,8]. 

In general, the literature is full of analytical methods that attempt to determine pyrimidines or their biosynthetic intermediates for differential diagnosis of hereditary metabolic diseases [9]. Our direct chromatographic separation involved a stationary phase HILIC column, as performed by other groups [10,11,12,13,14]. The proposed method allowed for the simultaneous determination of both OA and creatinine in a 10-min chromatographic run, presenting the advantage of expressing the result as a ratio of OA and creatinine concentration in urine through a single method. However, the specific aim of this work was not to develop an innovative chromatographic method for determining urinary OA. Instead, the aims of this study are as follows: (i) to quantify urinary OA in healthy subjects (males and females); (ii) to highlight the differences, in terms of OA concentration, among different sexes and age groups; (iii) to evaluate if OA level variations were related to organic or clinically detectable adaptations.

The determination of OA in urine has been the subject of attention by clinical laboratories, specifically when the phenotypic evidence of OA becomes part of genetic diseases. The classes of hereditary pathologies linked to an accumulation of OA are those that concern the biosynthesis defects of creatine, the metabolism of pyrimidines and the defects of the urea cycle (Figure 1). 

## 2. Methods

### 2.1. Materials and Analytes

OA, amino-OA (used as IS) and creatinine were purchased from Sigma-Aldrich (Milan, Italy). UHPLC-MS grade acetonitrile was purchased from Carlo Erba (Milan, Italy). UPLC-MS grade water was obtained from Thermo Fisher system (Milan, Italy). Formic acid and ammonium formate were purchased from Biosolve (Rome, Italy). Sodium hydroxide with purity ≥98.0% *w*/*w* was purchased from Sigma-Aldrich (Milan, Italy). The Sigma-Aldrich (Milan, Italy) synthetic negative urine control, “Sigmatrix urine diluent”, for blank sample (LEV0) preparation was also purchased from Sigma-Aldrich (Milan, Italy).

### 2.2. Standard and Quality Controls 

Stock solutions were used independently to spike blank urine to achieve 6 levels of calibration STD and 2 different QCs: high and low (QC H and L, respectively). Calibration and QC concentrations were as follows: 80.22 µmol/L (ULOQ), 41.79 µmol/L, 22.57 µmol/L, 5.27 µmol/L, 1.19 µmol/L and 0.15 µmol/L (LLOQ) for STD; 12.35 µmol/L (QC H) and 0.73 µmol/L (QC L) for QCs. Water UPLC-MS grade was used as a solvent for preparing creatinine and orotic acid stock solution. In particular, for orotic acid, the solution was basified with NaOH 0.1N. The concentration of creatinine calibration solution was 500 µmol/L (LLOQ), 1000 µmol/L, 1500 µmol/L, 2500 µmol/L, 5000 µmol/L, 10,000 µmol/L (ULOQ) and the QC concentration were 900 µmol/L (QC L) and 8500 µmol/L (QC H).

### 2.3. Standards, Quality Controls and Patients Samples Extraction

Prior to analysis, the STDs and QCs were kept at −4 °C for no more than 3 months, thus avoiding multiple freeze–thaw cycles. Urine samples from healthy volunteers were kept at −4 °C until the start of the extraction procedure. After thawing at room temperature, urine samples from healthy volunteers, STDs and QCs were all treated with the following procedure: 5 µL of the IS working solution were added to a volume of 50 µL of samples, STD and QCs, which were extracted with the use of 950 µL of acetonitrile.

The samples were vortex-mixed for at least 10 s. Subsequently, all samples were centrifuged at 2490× *g* for 10 min at 4 °C. 

Then, the supernatant (100 µL) were diluted with 400 µL of water:acetonitirle 80:20 *v*/*v* and moved to total recovery vials. Eventually, the volume of 10 µL was injected in the chromatographic system.

### 2.4. Chromatographic Conditions

The chromatographic system was an UPLC Nexera X2 ^®^ (Shimadzu, Milan, Italy) with a SIL-20AC^®^ auto-sampler, a binary pumps system as double LC-20AD, a column manager CTO-20AC^®^ column oven, DGU-305 degasser unit and CBM-20A as a controller. 

The flow rate was settled at 0.35 mL/min; the mobile phases used was 20 mM of ammonium formate at 0.2% *v*/*v* formic acid in water (Phase A) and 0.2% *v*/*v* formic acid in acetonitrile (Phase B); chromatographic elution was granted for satisfactory peaks separation.

The run duration lasted for 10 min. The autosampler was set at 4 °C. 

The van Deemter model was used to evaluate separation efficiency considering following parameters: *N* (theorical plates number) and HETP (*H*) (height of theorical plate). These parameters were included in the following equation: $N=16\cdot {\left(\frac{t_{R}}{W_{b}}\right)}^{2}$ and $H=\frac{L}{N}$ where *t_R_* was retention time expressed in minutes, *W_b_* was the width calculated at the base of the peek and *L* was the length of column in millimetres. These variables were watched for the regulation of the analytical procedure and derives were avoided.

### 2.5. Mass Spectrometry Conditions 

Tandem MS detection was carried out via a MS/MS Triple Quad 4500^®^, AB Sciex (Milan, Italy) with an ESI interface. The OA and related IS were detected in ESI negative ionization mode (ESI−), while creatinine was optimized in negative ionization (ESI−) (Table 1).

MS condition was optimized with an infusing reference solution for each analyte and relative IS (concentration of 1 ppm in aqueous acetonitrile) at 5.0 μL/min into the system. The infusion mode was set on “combined” to mix the reference solution with the flow 0.1 mL/min (Phase A and Phase B 50% *v*/*v*) from the LC instrument.

Nitrogen (>99.9%) produced with a LC-MS 40-1 nitrogen generator (Claind, Como, Italy) was used as nebulizing and heating gas, while argon was used as collision gas. 

The general conditions for positive ionization were electrospray voltage at 3.5 kV, source temperature at 550 °C and nebulizing gas flow at 1000 L/h.

### 2.6. Method Validation

According to the ICH Guidelines for Bioanalytical Method Validation, the analytical method was fully validated (see paragraph 7.1: “Methods for analytes which are also endogenous molecules” [15,16]) for the following parameters: specificity and selectivity, linearity range, LOD, LOQ, accuracy, intra-day and inter-day precision, robustness and matrix-effect (Table 2).

#### 2.6.1. Analytical Selectivity and Specificity

Selectivity is “the extent to which the method can be used to determine particular analytes in mixtures or matrices without interferences from other components of similar behavior” [17]. The selectivity was assessed by analyzing a blank sample (LEV0) obtained from 6 different lots of urine. The response detected was assessed through percent deviation (Δ%) from an LLOQ concentration level. These results were accepted with Δ% lower than 20% (absolute value) for analyte responses and ≤5% (absolute value) for IS responses. 

#### 2.6.2. Calibration 

The calibration consisted of 6 levels in addition to LEV0. The interpolation OF area/area IS and concentration was calculated using the least square method. A correlation factor of R^2^ > 0.995 was considered as the acceptance criteria and was linearity tested by Olivieri [18]. The Δ% of a single concentration level was calculated as follows: $∆\%=\frac{\left(C_{c}-C_{t}\right)}{C_{t}}\times 100$. The Δ% for each concentration level was accepted with values not exceeding ±20% for LLOQ and ±15% for other concentration levels [17].

The LOD was estimated using the Hubaux–Vos algorithm [19,20]. 

The LOQ was defined theoretically as three times of the LOD according to the IUPAC method. We calculated it experimentally starting from the Hubaux–Vos algorithm. The chromatogram of urine sample corresponding to 0.103 µmol/L (LOQ range) is reported in Figure 2.

#### 2.6.3. Repeatability and Reproducibility 

Reproducibility and repeatability were assessed with 10 replicates of QCs at 4 concentration levels in 3 sessions. The statistical analysis of this data was executed at 97.5% level of confidence of t-Student distribution.

In this context, we established intra-laboratory precision with limit repeatability calculus, as follows: $$r=\sqrt{2}\times t\times s_{r}$$
where *t* represented the t-student at $\left(1-\alpha \right)=0.975$ with 9 degrees of freedom for 10 experiments of intra-day repetition. 

The assessment of precision in repeatability and reproducibility conditions were evaluated by r $RSD\%=\frac{s}{\bar{x}}\times 100$, where s represented sR or reproducibility conditions and $\bar{x}$ is the mean value of 10 measures executed by single operator on the same sample. The repeatability measures were conducted on the same day while the reproducibility was measured on 2 different days by 2 different operators. 

#### 2.6.4. Precision Evaluation

In total, 10 replicates of QCs at 3 levels of concentration were analyzed in 3 different sessions by 2 different operators. The condition of acceptability for CV% was represented as follows: ±15% for each level, except for the LLOQ ± 20%. 

*R*(%) is expressed as $R\left(\%\right)=\frac{C_{f}-C_{s}}{C_{a}}\times 100.$ *C_f_* is the mean concentration of spiked QCs, *C_s_* is the mean concentration of not spiked QCs and Ca is the concentration of QCs.

Accuracy evaluation in accordance with the ICH guidelines [5] was performed through a bias % calculus as $b\left(\%\right)=\frac{\bar{x}-x_{ref}}{x_{ref}}\times 100$ where $\bar{x}$ represents the mean of the measurements and $x_{ref}$ was the reference value of proficiency test specimens (EQA).

The uncertainty of measurement was evaluated using the Horowitz heuristic model. The conditions of applicability of the Horwitz equation were verified on data distribution as follows: ratio between *s_r_* and *sR* comprised between 0.50 and 0.67. The *sR* obtained through the Horwitz equation was calculated by $sR=\frac{C}{100}\cdot 2^{\left(1-0.5\mathrm{Log}C\right)}$ where *C* was the concentration level of single analytes reported as mass ratio.

#### 2.6.5. Method Robustness ad Matrix Effect

Method robustness was assessed performing OA quantification in QCs at 3 levels of concentration. Further, 10 replicates of each level were processed in 2 analytical runs.

Interactions between different urine conditions and instrumental response were evaluated by response factor (*RF*) calculated as follow: $RF_{i}=\frac{A_{i}}{C_{i}}$, where Ai was the single analyte area and *C_i_* was the corresponding concentration. The deviations for QCs in each urine conditions for each analyte were evaluated with Δ*RF*% as follows: $\Delta RF\%=\frac{RF_{i}-\bar{RF}}{\bar{RF}}\times 100$: mean *RF* was calculated by relation $\bar{RF}=\frac{\sum_{i=1}^{n}RF_{i}}{n}$. In the last equation, *n* = 4 was concentration level of QCs. 

In precision evaluation, acceptance criteria were ΔCV% between ±15%, as recommended by ICH guidelines.

Furthermore, in accordance with ICH guideline, the matrix effect was evaluated by 3 replicates of QC1 and QC2 in 6 different urine lots. The analytes average response was compared to theorical concentration with percent deviation in an acceptability range of ±15% for ULOQ and ±20% for LLOQ. The same acceptability values were considered in CV% precision evaluation of response data.

#### 2.6.6. Stability 

Sample stability investigation was performed on QCs at 2 different concentrations to study the feasibility of sample collection. 

Considered timings for stability evaluation included 10, 30 and 90 days. Samples were stored both at −20 °C and −80 °C. 

Stability was expressed as the Δ% in each analyte concentration between freshly prepared samples and those kept at −80 °C and −20 °C.

Analyte freezing stability and thaw cycle (−80 °C) were calculated as deviation percentage (Δ%) in concentration for each analyte through the following equation: $\Delta \%=\frac{{[\bar{X]}}_{t0}-{[\bar{X]}}_{t1}}{{[\bar{X]}}_{t0}}\times 100$ where ${[\bar{X]}}_{t0}$ represents mean concentration value for each analyte in five replicates of each fresh $\left(t_{0}\right)$ QC, while ${[\bar{X]}}_{t1}$ stands for the mean value for each analyte in 5 replicates of each QC after every freeze-and-thaw cycle $\left(t_{1}\right)$. Δ% between ±15% was considered an acceptability condition. Further, 3 freeze-and-thaw cycles were performed. 

### 2.7. Enrollment Patients 

The clinical study to evaluate the physiological concentration of patients in good health has been approved by the Corporate Ethics Committee of the University Hospital “Città della Salute e della Scienza”.

The study is based on the following inclusion and exclusion criteria in accordance with ICH guideline E8 (R1) on general considerations for clinical studies [21].

Patient enrollment was authorized by ethical committee A/2.4.8. n.621/2020. 

Inclusion criteria was comprised of subjects aged between 0 and 70 years. Exclusion criteria is listed as follows:TabagismActive drug addiction, alcohol dependence and regular intake of psychodrugsTaking therapies within 30 days before the date of the estrogen and progestin collection even at low dosageRenal failureLiver failureSubjects undergoing transplantation of liver, kidneys, heart and lungsSubjects undergoing prosthetic limb surgery within the previous 24 monthsPeople with inherited metabolic disordersSepsi in progress or recentCoagulation disorders or thrombophlebitis in progressSubjects suffering from dyslipidemic diseases, dismetabolic or other known diabetesPresence of abdominal surgery in the last 24 monthsPregnant womenPresence of acute and/or chronic inflammatory or inflammatory pathology in the presence or absence of treatment with NSAIDs (including acetaminophen), glucocorticoids, immunomodulators and antibioticsTaking supplements based on vitamins (group B, C, D, E, niacin) and omega-3Presence of known cardiovascular, hepatic, renal and pneumological pathologiesPresence of oncological pathologies of any nature.

In order to be more precise in enrollment, and to avoid artifacts in the collected urine samples, a questionnaire was administered concerning the habits of life and in particular the intake of certain foods (e.g., crustaceans) in the 24 h before urine sampling that could affect the orotic aciduria.

## 3. Results

Our results are in accordance with ICH criteria. They are acceptable results for intra-day and inter-day precision, as well as for the matrix effect. They also show good linearity and ideal accuracy. 

Analytes OA and creatinine were identified and quantified with high-performance parameters of repeatability, reproducibility, robustness, precision and accuracy. Creatinine has been previously determined to normalize the concentrations of OA with homogenous and representative measures of renal clearance. The quantification method was based on the IS approach for signal and matrix effect suppression. Figure 3 reports the cumulative chromatogram of OA, IS and creatinine Analytes were identified by two MRM transitions with S/N > 50 in LOD range. The analytical method clearly distinguishes between urine specimens from the normal and pathological patients at 97.5% of level of confidence.

Here, we collected urine from 66 healthy volunteers (29 males and 37 females). The median age for females was 35.44 years (IQR: 7.12–47.01 years), ranging from 2.76 years (IQR: 2.76–6.68 years) from 0 to 10 years (N = 11), 11.87 years (IQR: 11.87–22.98 years) from 10 to 30 years (N = 7), 35.96 years (IQR: 35.96–45.53 years) from 30 to 50 years (N = 11) and 55.72 years (IQR: 52.63–59.15 years) from 50 to 70 years (N = 8). The median age for males was 14.84 years (IQR: 7.13–46.55 years), ranginf from 2.10 years (IQR: 0.58–4.67 years) from 0 to 10 years (N = 8), 12.79 years (IQR: 11.65–19.04 years) from 10 to 30 years (N = 9), 48.75 years (IQR: 39.73–56.27 years) from 30 to 70 years (N = 12). 

The analysis of oroticuria data in males and females showed, at a confidence level of 95% (α = 0.05), a significant difference (${\left.p_{value}\right|}_{\nu =18;\alpha =0.05}=0.0082$, *p* < 0.05) only in the first decade of life, where the oroticuria of males was about double that of females (Table 3). 

In other age ranges, we saw no significant difference, even if, considering the dispersion of the data, there was certainly a trend towards higher values of oroticuria in females of childbearing age compared to the minimum and maximum values of male oroticuria in the age group between 30 and 50 years (Figure 4 and Figure 5).

## 4. Discussion

OA is an intermediate in pyrimidine nucleotide biosynthesis. It is a component of mammalian milk and has also been recognized as a vitamin and product that helps reinforce muscles. OA is abundant in serum and urine of patients with hereditary orotic aciduria or enzyme defects in the urea cycle. The development and optimization of sensitive techniques to measure orotate concentration in biological samples, without other compounds interference, was prompted by studies on dairy products, as well as the discovery of OA in human fluids caused by inborn errors of pyrimidine metabolism [22,23,24,25,26,27,28]. In studies on the development of HPLC methodology, earlier microbiological, enzymatic, polarographic and colorimetric experiments were carefully considered [29,30,31]. Since DO and other pyrimidine metabolites lacking a double bond were not taken into consideration for analysis, it is clear that HPLC detection systems were primarily dependent on the distinctive UV absorbance of OA and orotidine. The isotope dilution method [32] and further advanced technology, such as differential pulse polarography [33], 1H-NMR spectroscopy [34], capillary zone electrophoresis [35] and fluorescence sensing [36], received little application in clinical chemistry. GC-MS was used to screen a sizable population of patients (N = 7500) for organic acids, including high OA [37,38]. DO and CA or dihydrouracil from pyrimidine catabolism was simultaneously determined using HPLC-MS/MS for routine clinical analysis, replacing the previously required chemical derivatization process for metabolites in samples for highly sensitive fluorometric detection [39]. The van Deemter model is simplified when we consider gradient elution, but the HETP and N quantities are useful parameters to verify the drift of the method and the loss in time of chromatographic column separative efficiency [40].

Here, we aimed to suggest a methodological chromatographic method for the determination of oroticuria in the urine of healthy patients. Moreover, we evaluated the sex and age-related different expression of pyrimidine intermediate biosynthesis. It is crucial to try to understand factors associated with the physiological urine profile of OA during life, in addition to the methodological and descriptive features of oroticuria in males and females. The comparison of the analyzed data showed a significant difference between females of pediatric and prepubescent age (0–10 years) and males, which showed about twice the OA levels than females. This tendency is soon reversed in the years just before menopause. Males, on the other hand, show an almost constant trend up to 70 years of age. The hypotheses we can now make regarding the biochemical mechanisms that determine such variations in the excretion of OA are various. In particular, it should be speculated that the influence of prolactin and luteinizing hormones levels can determine a different expression of orotate dehydrogenase [41,42], and consequently a biosynthetic cascade of pyrimidines control. In this way, we can explain the oroticuria increase in the prepuberal period (both for males and females) and the relative deviation (only for females) in the pre-menopausal period. 

A significant knowledge gap still exists in the area of sex differences in clinical practice. Given the investigation of the safety profile and treatment outcomes, it is still uncommon for female animals and humans to be included in research. However, sex-based major disparities in different analytes kinetics and dynamics have been documented over the past decade. Males and females differ specifically in terms of absorption, distribution, metabolism and clearance mechanisms [43]. Sex differences could be due to hormone-affected urine protein levels and gastrointestinal motility, which influence protein binding and drug absorption. High progesterone levels during the luteal phase modify absorption by relaxing smooth muscles and lengthening small intestine retention time [43]. The volume of distribution, half-life and free drug concentration, are all affected by an increase in 1 acid glycoprotein levels [44,45,46]. The menstrual cycle fluctuating hormone levels could also affect the hepatic enzymes that regulate metabolism. Contrary to androgens, progesterone and estrogens suppress liver microsomal enzymes. Progesterone could, however, also increase the activity of hepatic enzymes [45,47,48,49,50]. Eventually, CYP3A4, CYP2A6 and CYP2B6 exhibit higher activity in women than in males; nevertheless, CYP1A2, CYP2E1 and UGT show decreased activity in women, which is sex-related [51]. Differences in body weight may have an impact on volume of distribution and clearance. Additionally, in women, it was reported that lower glomerular filtration rate and higher body fat percentage affected compound elimination.

Our data confirms the need to consider the cyclical nature of female hormone expression as a possible factor that influences the trend of any molecule. In general, the male system presents a much more uniform trend over the course of life. This difference must always be considered from a biochemical point of view, with potential repercussions in clinical diagnosis, therapy and patient care. Our research highlights the different oroticuria between the male and female population.

## 5. Conclusions

The described analytical method allows for the accurate and precise dosing of urine OA and the determination of creatinine for OA concentration normalization. This method could also be easily used in the clinical routine. Moreover, the analysis of the biochemical spectrum of hereditary pathologies linked to an accumulation of OA, the bio-synthesis defects of creatine, the metabolism of pyrimidines and the defects of the urea cycle, are useful in determining the clinical and biochemical pattern of these diseases.

The obtained results suggest that a gender and tailored approach should be a useful tool for personalized medicine in clinical practice. This is a pilot study that needs further investigation by expanding the cohort of patients and investigating the expression of other metabolites and hormones. Moreover, data including male and female hormonal phases, body weight and body fat distribution are indispensable in clarifying the obtained results. Eventually, the evaluation of single nucleotide polymorphisms in genes involved in the biosynthesis, metabolism and clinical consequences of OA, creatinine and pyrimidine are necessary for a better interpretation of kinetic and clinical results.

## Figures and Tables

**Figure 1 jpm-13-01443-f001:**
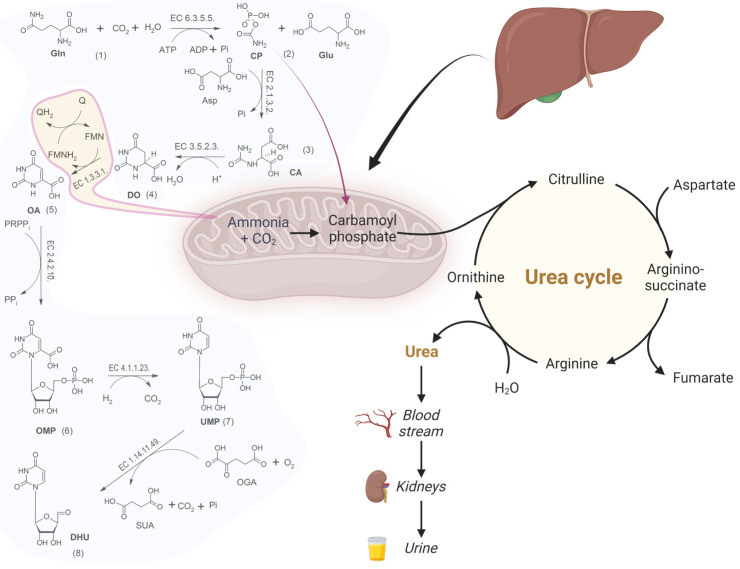
Biosynthesis of pyrimidine and urea cycle. Gln (glutamine), CP (carbamoylphosphate), Glu (glutamic acid), Asp (aspartic acid), CA (carbamoylaspartate), DO (dihydro-orotic acid), OA (orotic acid), FMN (Flavinmononucleotide), FMNH2 (Flavinmononucleotide reduced), OMP (orotidine-5′- monophosphate), UMP (uridine—5′—monophosphate), DHU (dihydrouridine), OGA (2-oxoglutaric acid), SUA (succinic acid), ATP (adenosintriphosphate), ADP (adenosindiphosphate), PPi (pyrophosphate), Pi (phosphate).

**Figure 2 jpm-13-01443-f002:**
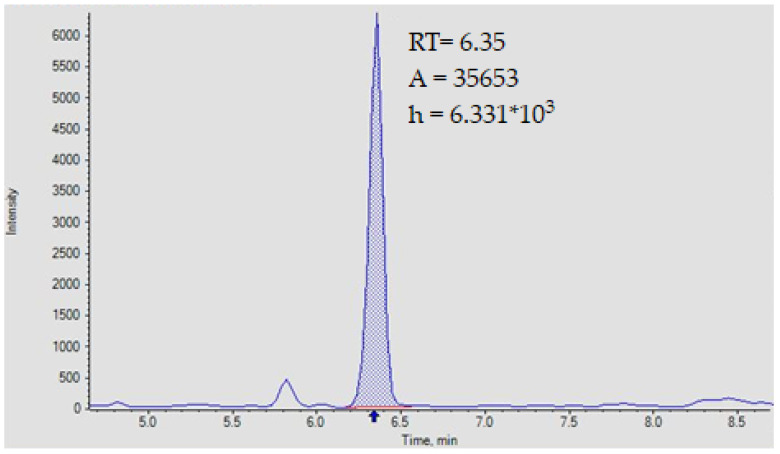
Chromatogram of orotic acid corresponding to LOQ range concentration. RT, retention time; A, area of peak; h, height of peak.

**Figure 3 jpm-13-01443-f003:**
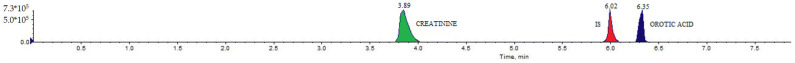
Cumulative chromatogram of orotic acid, its relative internal standard and creatinine.

**Figure 4 jpm-13-01443-f004:**
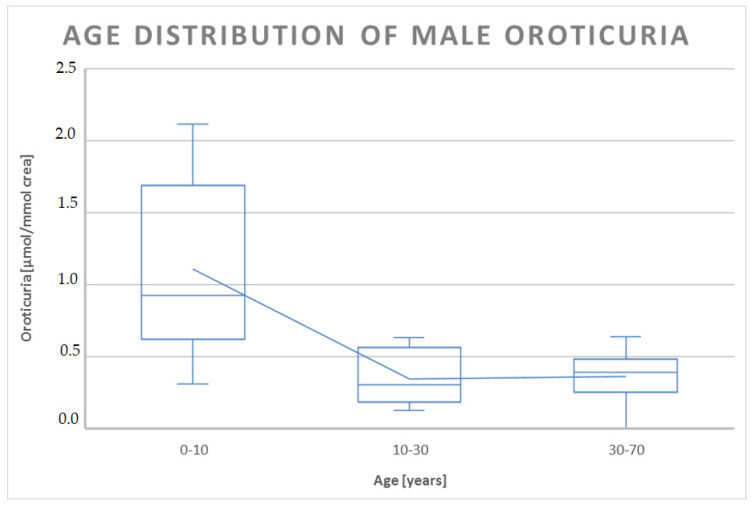
Boxplot representation of the distribution of oroticuria according to the age range of male patients enrolled in the study. Oroticuria values are expressed in µmol/mmol of creatinine.

**Figure 5 jpm-13-01443-f005:**
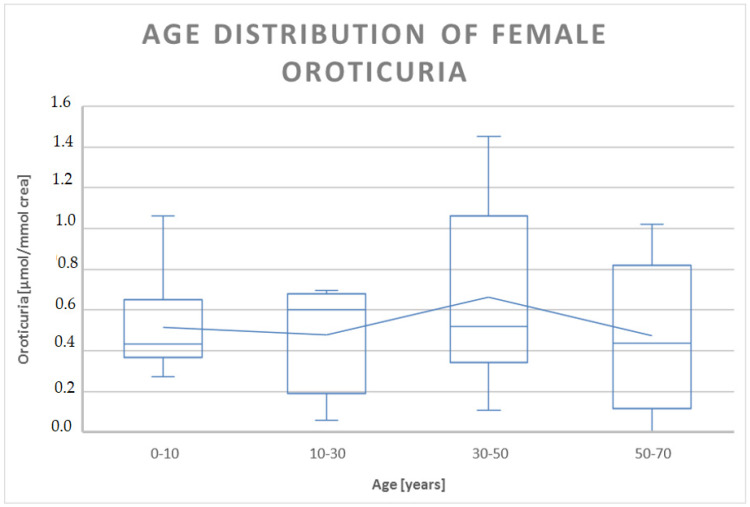
Boxplot representation of the distribution of oroticuria according to the age range of female patients enrolled in the study. Oroticuria values are expressed in µmol/mmol of creatinine.

**Table 1 jpm-13-01443-t001:** MRM transition of analytes and corresponding internal standard.

Analyte/Is, Esi+/−	Ion Quantifier MRM (M/Z)	Ion Qualifier MRM (M/Z)	Declustering Potential (V)	Entrance Potential (V)	Collision Energy (V)	Collision Cell Exit Potential (V)
Orotic Acid (−)	155→111		−58.74	−9.51	−13.34	−14
		155→42	−58.47	−9.51	−47.82	−12.3
Amino-Orotic Acid (−)	170→126		−64.96	−14	−17.09	−14.09
		170→83	−64.96	−14	−27.08	−18.97
Creatinine (+)	114→86		130.6	10	16.68	12.09
		114→58	130.6	10	19.73	15.96

**Table 2 jpm-13-01443-t002:** Considered parameters for method analytical validation.

	Orotic Acid			Creatinine		
	LEV 1	LEV 2	LEV 3	LEV 1	LEV 2	LEV 3
Mean Value of Calculated Concentration In µmol (Xm)	0.73	1.81	12.35	600.0	2800.0	6500.0
Standard Deviation on Repeatability (sr)	0.07	0.33	0.19	102.6	287.7	268.1
Confidence Interval (IF)	1.22	3.05	20.83	1011.5	4720.4	10958.1
Repeatability Coefficient (r)	0.27	1.35	0.77	423.7	1188.0	1107.0
Relative Standard Deviation on Repeatability (RSDR)	9.09	18.03	1.50	17.1	10.3	4.1
Relative Standard Deviation on Reproducibility (RSDR)	16.79	14.64	10.96	6.1	4.8	4.3
Determination coefficient of Calibration Curve (R^2^)	0.999			0.998		
Limit of Quantification (LOD)	0.68			0.046		
Limit of Detection (LOQ)	1.49			0.100		
Recovery Percentage (R(%))	96.14			92.00		

**Table 3 jpm-13-01443-t003:** Data summary on the sample of male (upper panel) and female (lower panel) healthy volunteers enrolled. Oroticuria values are expressed in moles of orotic acid.

**Males**	**Year Range**	**0–10**	**10–30**	**30–70**	
	Min	0.31	0.13	0.00	
	Max	2.11	0.63	0.64	
	SE	0.23	0.06	0.05	
	s	0.65	0.19	0.17	
	X_m_	1.11	0.35	0.36	
	n	8	9	12	
**Females**	**Year range**	**0–10**	**10–30**	**30–50**	**50–70**
	Min	0.27	0.06	0.11	0.00
	Max	1.06	0.69	1.45	1.02
	SE	0.07	0.11	0.12	0.15
	s	0.23	0.27	0.43	0.39
	X_m_	0.51	0.48	0.66	0.47
	n	12	6	12	7

## Data Availability

The data that support the findings of this study are not openly available due to reasons of sensitivity and are available from the corresponding author upon reasonable request.

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
