# Peer review of "The Strange Case of Orotic Acid: The Different Expression of Pyrimidines Biosynthesis in Healthy Males and Females"

_jpm, 2023, doi:10.3390/jpm13101443_

Round 1

Reviewer 1 Report

The paper describes a method for the determination of orotic acid in urine with LC-MS/MS. It describes not only the method but also a significant difference in the concentration of orotic acid between the female and male population of children under 10 years of age.  

Even though this paper reveals significant gender-related differences in orotic acid urine concentration, unfortunately, it suffers from many drawbacks which prevent this paper from being published in its present form.  

Firstly, there is no significant novelty in the LC-MS/MS method described in this paper compared to the literature. There are multiple chromatographic methods already published from 1980 onwards that can provide measurements of orotic acid in urine. If there is any novelty, it should be clearly presented. Furthermore, there is hardly any discussion in terms of critical comparison between this method and all other methods already published for this analyte.  

Secondly, the validation guidelines used for assessing the appropriateness of this analytical method are used for drugs in biological fluids. Orotic acid is here analyzed as an endogenous analyte. Therefore, more appropriate would be the CLSI guidelines for this purpose.  

Thirdly, the number of individuals in each age group is too small to adequately represent the general population, and to allow the confident conclusion of the sex-related differences in the first group, especially considering the large variation observed in both males and females in different age groups.  

Fourthly, the calibrated range seems too high compared to the values obtained  

Fifthly, how was the analyte calibrated in urine – which blank analyte-free matrix was used for this purpose?  

Sixthly, in the presented Figure 1, in the reaction catalyzed by the dihydroorotate dehydrogenase, there is no O2 as an electron acceptor, but it should be NAD+.  

Seventhly, the way the lower limit of quantitation was established, by multiplying the LOD by a factor of three is only appropriate for rough estimation, there is no proof that the method   

Eighthly, there is no chromatogram presented for the limit of quantitation nor for any of the study samples.  

Ninthly, the Van Deemter model for the presentation of the column separation efficiency in gradient methods is not appropriate – peak capacity should be used instead.  

Lastly, there are a number of mistakes that are difficult to forgive, such as wrong units (moles of orotic acid instead of micromoles per mmol of creatinine in Table 4), several mistakes of writing about the drug instead of orotic acid, and plasma as the matrix instead of urine. Where and how did the authors o obtain lipemic and haemolytic urine? 

There are number of mistakes. The manuscript should be subjected to careful proofreading.

Author Response

Turin, September 2023

Dear Editor,

Please, find enclosed a revised manuscript (Manuscript ID: jpm-2569344) to be considered for publication in Journal of Personalized Medicine, Special Issue on " Cancer Genetics and Clinical Research” as Article. As follow you can find the point-to-point response to reviewer questions. In addition, the uploaded version of the manuscript contains the highlighted correction (in yellow).

We hope that you will find our data worth of attention for Journal of Personalized Medicine readers.

Best regards,

Sarah Allegra

REVIEWER 1

The paper describes a method for the determination of orotic acid in urine with LC-MS/MS. It describes not only the method but also a significant difference in the concentration of orotic acid between the female and male population of children under 10 years of age. 

Even though this paper reveals significant gender-related differences in orotic acid urine concentration, unfortunately, it suffers from many drawbacks which prevent this paper from being published in its present form. 

Dear Reviewer, thank you very much for the thorough review. We agree to all specific comments addressed and have revised our paper in light of the useful suggestions. Answers to the specific comments/suggestions/queries are as follows.

Firstly, there is no significant novelty in the LC-MS/MS method described in this paper compared to the literature. There are multiple chromatographic methods already published from 1980 onwards that can provide measurements of orotic acid in urine. If there is any novelty, it should be clearly presented. Furthermore, there is hardly any discussion in terms of critical comparison between this method and all other methods already published for this analyte. 

Thank you for your professional comment. It is true that the determination of orotic acid in urine has already been implemented through RP separation chromatographic methods since the 1980s. The chromatographic separation performed by us is of a direct type using a column with stationary phase HILIC, as other groups have done since 2010 (https://doi.org/10.1002/jssc.200900758; 10.1016/j.chroma.2011.10.086; 10.1007/s00216-019-01661-0; 10.1016/j.jpba.2020.113551; 10.1016/j.chroma.2023.464065); this clarification has been inserted in the text, thank you for the suggestion. However, the method allows both orotic acid and creatinine to be determined simultaneously in a 10-minute chromatographic run, presenting the advantage of expressing the result as a ratio of the concentration of orotic acid and creatinine in the urine by a single method. In addition, the purpose of the work is not methodological: we aim to quantify urinary orotic acid in healthy subjects (males and females) of different ages; we highlight the aim of the study in lines 108-117.

Secondly, the validation guidelines used for assessing the appropriateness of this analytical method are used for drugs in biological fluids. Orotic acid is here analyzed as an endogenous analyte. Therefore, more appropriate would be the CLSI guidelines for this purpose. 

Thank you for raising this point of potential confusion. The guidelines "ICH M10 on Bioanalytical method validation and study sample analysis" include the validation of chromatographic analytical methods for endogenous substances, for which there is also a dedicated paragraph, precisely 7.1 "Methods for analytes which are also endogenous molecules" (25 July 2022 EMA/CHMP/ICH/172948/2019 Committee for Medicinal Products for Human Use, ICH guideline M10 on bioanalytical method validation and study sample analysis; https://doi.org/10.1208/s12248-019-0398-7), as addended in the text (lines 194-195).

Thirdly, the number of individuals in each age group is too small to adequately represent the general population, and to allow the confident conclusion of the sex-related differences in the first group, especially considering the large variation observed in both males and females in different age groups. 

Thank you for your comment. We collected urine from 66 healthy volunteers, 29 males and 37 females; this is a pilot study and therefore the conclusion to be drawn can be verified by expanding the cohort of patients and investigating the expression of other metabolites and hormones. This study limit has been inserted in the conclusion section.

Fourthly, the calibrated range seems too high compared to the values obtained 

Thank you for your revision. The measuring range is large but the calibration curve has good performance even at low concentration values as shown by the precision data on the concentration level 1 in Table 2.

Fifthly, how was the analyte calibrated in urine – which blank analyte-free matrix was used for this purpose? 

Thank you for your interest. The white sample consists of artificial urine (Sigmatrix urine diluent - Synthetic negative urine control) purchased from Sigma Aldrich. This indication was given in lines 136-138.

Sixthly, in the presented Figure 1, in the reaction catalyzed by the dihydroorotate dehydrogenase, there is no O2 as an electron acceptor, but it should be NAD+. 

Thank you for raising this point of potential confusion. The dihydroorotate dehydrogenase i Class 1 family uses flavin mononucleotide (FMN) as a cofactor which undergoes an oxidation/reduction cycle coupled to several oxidants such as oxygen, fumarate or ubiquinone present in the cytosol. In this case, to be precise, the electron acceptor is not NAD+ (which is used by dihydroorotate class 2 dehydrogenase) but ubiquinone (https://doi.org/10.1016/s0969-2126(00)00077-0; 10.1146/annurev.bi.49.070180.001345). Figure 1 has therefore been corrected.

Seventhly, the way the lower limit of quantitation was established, by multiplying the LOD by a factor of three is only appropriate for rough estimation, there is no proof that the method  

Thank you for your comment. The LOD for orotic has been shown showed in the new Table 2, estimated by the IUPAC method (). It was corrected with the Hubaux-Vos algorithm, as reported in lines 221-223.

Eighthly, there is no chromatogram presented for the limit of quantitation nor for any of the study samples. 

Thank you for your suggestion. The new Figure 2 has been added with the chromatogram for urine sample with orotic acid concentration of 0.103 µmol/L, corresponding to the LOQ (lines 220-221).

Ninthly, the Van Deemter model for the presentation of the column separation efficiency in gradient methods is not appropriate – peak capacity should be used instead. 

Thank you for your revision. We agree with you, the Van Deemter model is simplified when we consider gradient elution, but the HETP and N quantities are useful parameters to verify the drift of the method and the loss in time of the separative efficiency of the chromatographic column. For example, by constructing a Shewhart control card, both HETP and N can be monitored over time by placing attention levels, and in this way, replace the column before the analytical process drift occurs. They are therefore indispensable parameters in the evaluation of reproducibility tests and new method verification at a distance of time, as clarified in conclusion section.

Lastly, there are a number of mistakes that are difficult to forgive, such as wrong units (moles of orotic acid instead of micromoles per mmol of creatinine in Table 4), several mistakes of writing about the drug instead of orotic acid, and plasma as the matrix instead of urine. Where and how did the authors o obtain lipemic and haemolytic urine?  

We apologize for the omissions or some errors that were present in the text. We have completely revised the article trying to eliminate all the errors and misunderstandings present.

Reviewer 2 Report

The manuscript entitled: The Strange Case of Orotic Acid: the divergent expression of  pyrimidines biosynthesis in heathy males and females, presents an interesting and robust method for the determination of oroticuria in urine of healthy patients, in addition to the sex-related different expression of pyrimidine intermediate biosynthesis. However, the following points should be addressed before further steps:

Title and abstract:

-while the title is interesting, and the abstract is informative, yet it is not easy to the reader to establish a quick relationship between the title and the abstract. It is advised to align the abstract with the title.

Methods

-Line 90: …..(each sample was treated as follows). describe in details what are the samples and how they were obtained.

Results:

-Is it possible to combine tables 3 and 4.

References:

-While still relevant, some references like No 27 is very old. please updated when possible?

-Abbreviation section needed

-Line 234: amend (This our results…).

Author Response

Turin, September, 2023

Dear Editor,

Please, find enclosed a revised manuscript (Manuscript ID: jpm-2569344) to be considered for publication in Journal of Personalized Medicine, Special Issue on " Cancer Genetics and Clinical Research” as Article. As follow you can find the point-to-point response to reviewer questions. In addition, the uploaded version of the manuscript contains the highlighted correction (in yellow).

We hope that you will find our data worth of attention for Journal of Personalized Medicine readers.

Best regards,

Sarah Allegra

REVIEWER 2

he manuscript entitled: The Strange Case of Orotic Acid: the divergent expression of  pyrimidines biosynthesis in heathy males and females, presents an interesting and robust method for the determination of oroticuria in urine of healthy patients, in addition to the sex-related different expression of pyrimidine intermediate biosynthesis. However, the following points should be addressed before further steps:

Dear Reviewer, thank you very much for the thorough review. We agree to all specific comments addressed and have revised our paper in light of the useful suggestions. Answers to the specific comments/suggestions/queries are as follows.

Title and abstract:

-while the title is interesting, and the abstract is informative, yet it is not easy to the reader to establish a quick relationship between the title and the abstract. It is advised to align the abstract with the title.

 Thank you for raising this point of potential confusion. Title and abstract has been modified, following your useful suggestions.

Methods

-Line 90: …..(each sample was treated as follows). describe in details what are the samples and how they were obtained.

 Thank you for your recommendation, in lines 148-152 a suggested description has been added.

Results:

-Is it possible to combine tables 3 and 4.

 Thank you for your suggestion, tables 3 and 4 has now been merged in a single table 3.

References:

-While still relevant, some references like No 27 is very old. please updated when possible?

As you recommended, a complete revision of the reference list has been performed. Thank you. 

-Abbreviation section needed

Thank you for your comment and suggestions. The list of abbreviation has been added in lines 37-91.

Reviewer 3 Report

The manuscript "The Strange Case of Orotic Acid: the divergent expression of 2 pyrimidines biosynthesis in heathy males and females" is devoted to a very important metabolite - orotic acid and its analysis in patients. The article is written very well, the design of the study is well planned. However, there are a number of questions:

1. The article does not provide a detailed description of the groups of patients that were examined. It would be nice to insert this information into the text of the article or into SI. The authors themselves indicate that the difference in the concentrations of orotic acid is different depending on age.

2. Also, in the form of information, it is necessary to add to the introduction what modern methods of detecting orotic acid are currently used in medicine (it is advisable to give several references). And in the discussion of the results, this information can be presented in more detail. This will make it much more interesting for readers.

3. It would be very good to present the biochemical spectrum of standard analytes of biochemical analysis and liver tests in the experimental group of patients, if only because the journal in which the article is planned to be published is also devoted to personalized medicine.

4. The genotyping of patients is very important, since the levels of orotic acid are highly dependent on the work of the genes of the xenobiotic detoxification system (cytochrome genes). For example, the activity of cytochrome P450 greatly affects the concentration of various metabolites in the patient's blood. Did the authors carry out such genotyping of the experimental group of patients?

Author Response

Turin, September, 2023

Dear Editor,

Please, find enclosed a revised manuscript (Manuscript ID: jpm-2569344) to be considered for publication in Journal of Personalized Medicine, Special Issue on " Cancer Genetics and Clinical Research” as Article. As follow you can find the point-to-point response to reviewer questions. In addition, the uploaded version of the manuscript contains the highlighted correction (in yellow).

We hope that you will find our data worth of attention for Journal of Personalized Medicine readers.

Best regards,

Sarah Allegra

REVIEWER 3

The manuscript "The Strange Case of Orotic Acid: the divergent expression of 2 pyrimidines biosynthesis in heathy males and females" is devoted to a very important metabolite - orotic acid and its analysis in patients. The article is written very well, the design of the study is well planned. However, there are a number of questions:

Dear Reviewer, thank you very much for the thorough review. We agree to all specific comments addressed and have revised our paper in light of the useful suggestions. Answers to the specific comments/suggestions/queries are as follows.

  1. The article does not provide a detailed description of the groups of patients that were examined. It would be nice to insert this information into the text of the article or into SI. The authors themselves indicate that the difference in the concentrations of orotic acid is different depending on age.

Thank you for your recommendation, in lines 307-314 a detailed description of the enrolled healthy volunteers has been inserted.

  1. Also, in the form of information, it is necessary to add to the introduction what modern methods of detecting orotic acid are currently used in medicine (it is advisable to give several references). And in the discussion of the results, this information can be presented in more detail. This will make it much more interesting for readers.

Thank you for your professional comment. The determination of orotic acid in urine has already been implemented through RP separation chromatographic methods since the 1980s. The chromatographic separation performed by us is of a direct type using a column with stationary phase HILIC, as other groups have done since 2010 (https://doi.org/10.1002/jssc.200900758; 10.1016/j.chroma.2011.10.086; 10.1007/s00216-019-01661-0; 10.1016/j.jpba.2020.113551; 10.1016/j.chroma.2023.464065); this clarification has been inserted in the text, thank you for the suggestion. In addition, the proposed method allows both orotic acid and creatinine to be determined simultaneously in a 10-minute chromatographic run, presenting the advantage of expressing the result as a ratio of the concentration of orotic acid and creatinine in the urine by a single method. Eventually, the aim of the work is not methodological: we quantified urinary orotic acid in healthy subjects (males and females) of different ages; we highlight the aim of the study in lines 108-117.

  1. It would be very good to present the biochemical spectrum of standard analytes of biochemical analysis and liver tests in the experimental group of patients, if only because the journal in which the article is planned to be published is also devoted to personalized medicine.

Thank you very much for your very interesting comment. The evaluation, the comparison and the discussion of biochemical spectrum of the analysis of the biochemical spectrum of hereditary pathologies linked to an accumulation of orotic acid, the biosynthesis defects of creatine, the metabolism of pyrimidines and the defects of the urea cycle fall outside of the aim of the proposed work; actually, it will be the focus of another our study, which include not only healthy volunteers. This information has been added in conclusion section (lines 408-411).

  1. The genotyping of patients is very important, since the levels of orotic acid are highly dependent on the work of the genes of the xenobiotic detoxification system (cytochrome genes). For example, the activity of cytochrome P450 greatly affects the concentration of various metabolites in the patient's blood. Did the authors carry out such genotyping of the experimental group of patients?

Thank you for your suggestion; unfortunately, the analysis of allelic discrimination for genetic variant in enzymes involved in the biosynthesis, metabolism and clinical effect of orotic acid, creatinine and pyrimidine have not been performed. Sure, we strongly agree with you: the genetic tests are necessary for a better interpretation of kinetic and clinical results (lines 417-419).

Round 2

Reviewer 1 Report

Review of the revision of the manuscript entitled »The strange case of orotic acid: the different expression of pyrimidines biosynthesis in heathy males and females«

Major

The chromatographic conditions are not adequately described (gradient, column data).  The chromatogram of creatinine and IS is missing. There is no critical assessment of the sample size when comparing the age and sex differentiated groups. The sample size is one of the most critical and weak spots of the presented study, so there should be adequate and strong discussion regarding the observed differences between males and females at the age 0-10. Especially, since there is no evident or logical explanation given in the discussion for this discrepancy.

Specific comments

Line 80: R2 is determination coefficient, not correlation factor of calibration curve

Line 89: ULOQ is usually not the upper level of quantification, but upper limit of quantification.

Line 134: UPLC grade acetonitrile is not a product of Carlo Elba; that manufacturer has UHPLC-MS grade acetonitrile. Remember, there is an important difference between the terms »UPLC« and »UHPLC«. The first is a trademark of Waters corporation, the second is a generic term that applies to this type of instruments from any vendor, Shimadzu, Agilent, Waters, Knauer, Thermo, etc.

Lines 140-145: what were the concentrations of stock solutions and what kind of solvents were used for their preparation (applies both to orotic acid and creatinine)? What were the final concentrations of creatinine in calibration and QC samples?

Line 159. Shimadzu does not produce an UPLC. Please see my previous comment regarding that issue (line 134).

Line 161. The patient enrolment and Ethical committee grant have nothing to do with the title of that heading »chromatographic conditions«. On the other hand, some very important data IS missing: the column model, dimensions, the mobile phase composition at various time points or the gradient etc.

Line 161. Which were the inclusion and exclusion criteria for the subjects?

Line 172. Explain the sentence containing the phrase »..and derives was prevented«.

Line 186. Please explain how was the flow combined from the chromatographic system with the flow of the infusion containing the reference standards. What is meant by the term »medium concentrations of phases«?

Line 197. The text refers to Table 2, which is missing in the manuscript.

Line 199-202: Table 1 – please add accuracy data.

Line 199-202, Table 1: How is LEV1 related to QC L?

Line 199-202, Table 1 and Lines 193-195: In the cited literature and guidelines, there is no reference for standard deviation on repeatability sR, repeatability coefficient R, nor for Rsdr and RSDR.

Line 199-202, Table 1: Which concentration level does the calculated recovery correspond to, QC-low, medium or high?

Line 208. Rephrase the sentence containing the unusual wording »gap minus«.

Line 219. The Hubaux-Vos algorithm is not a part of ICH-M10 guidelines which the authors claim to have followed for their validation of the method.

Line 155: Please report the centrifugal force in correct terms (g-units, not RPM).

Line 226, Figure 2: Please present also the chromatogram of internal standard and creatinine.

Line 244: What are the levels for spiking (the data on QC-M is missing)?

Line 265: How was the method's robustness tested and where are the results?

Line 262-276: The cited ICH M10 guidelines do not propose the calculation of response factors for matrix effect evaluation, but the preparation of low and high QCs in blank matrix from 6 individuals. Therefore, the unspiked response should be subtracted from the spiked response, thereby obtaining the signal gained from spiking in each lot of matrix. In effect, this is similar to the response factor calculation, nevertheless it is better to strictly adhere to the guidelines if they are reported to be followed.

Line 281: Consider the term sample stability instead of shelf life which is used for pharmaceutical and food industry products.

Line 298: Where are the results from matrix effect study?

Line 324, table 3: what were the levels of creatinine in each group?

Line 330: figures 3 and 4, and table 3: Why were 30-50 and 50-70 groups combined in the male samples?

Lines 383-392: The authors should carefully re-evaluate what to include in their discussion regarding the differences in the absorption and metabolism in children in the ages from 0-10 years and the effects of menstrual cycle. Especially considering that male subjects 0-10 years had significantly higher levels of OA compared to other groups.

Line 360: The inserted explanation of the Van Deemter model does not fit into the context of comparison between the presented method and other published methods, unless the authors conclude this is a crucial difference or benefit relative to other methods.

The manuscript should be subjected to careful proofreading by a native English speaker.

Author Response

Turin, September 2023

Dear Editor,

Please, find enclosed a revised manuscript (Manuscript ID: jpm-2569344) to be considered for publication in Journal of Personalized Medicine, Special Issue on " Cancer Genetics and Clinical Research” as Article. As follow you can find the point-to-point response to reviewer questions. In addition, the uploaded version of the manuscript contains the highlighted correction (in yellow).

We hope that you will find our data worth of attention for Journal of Personalized Medicine readers.

Best regards,

Francesco Chiara

Review of the revision of the manuscript entitled »The strange case of orotic acid: the different expression of pyrimidines biosynthesis in heathy males and females«

 Dear Reviewer, thank you very much for the thorough review. We agree to all specific comments addressed and have revised our paper in light of the useful suggestions. Answers to the specific comments/suggestions/queries are as follows.

Major

The chromatographic conditions are not adequately described (gradient, column data).  The chromatogram of creatinine and IS is missing. There is no critical assessment of the sample size when comparing the age and sex differentiated groups. The sample size is one of the most critical and weak spots of the presented study, so there should be adequate and strong discussion regarding the observed differences between males and females at the age 0-10. Especially, since there is no evident or logical explanation given in the discussion for this discrepancy.

Specific comments

Line 80: R2 is determination coefficient, not correlation factor of calibration curve

Line 89: ULOQ is usually not the upper level of quantification, but upper limit of quantification.

Thank you for your revision, the suggested corrections have been made.

Line 134: UPLC grade acetonitrile is not a product of Carlo Elba; that manufacturer has UHPLC-MS grade acetonitrile. Remember, there is an important difference between the terms »UPLC« and »UHPLC«. The first is a trademark of Waters corporation, the second is a generic term that applies to this type of instruments from any vendor, Shimadzu, Agilent, Waters, Knauer, Thermo, etc.

Thank you for your interest. The text has been changed as you suggested.

Lines 140-145: what were the concentrations of stock solutions and what kind of solvents were used for their preparation (applies both to orotic acid and creatinine)? What were the final concentrations of creatinine in calibration and QC samples?

Thank you for your comment. Although the requested information is of summary importance for the aim of the article, the section has been integrated in lines 147-152.

Line 159. Shimadzu does not produce an UPLC. Please see my previous comment regarding that issue (line 134).

Thank you for your revision. The UPLC acronym is synonymous with UHPLC from a technical and non-commercial point of view. In particular it indicates the ability of the instrument (pumps, autosampler and hydraulic circuit) to reach at least 900bar of maximum pressure. This is so that columns with a stationary phase of less than 2.0um can be used with flows of 0.3 - 0.6 ml/min. That said Shimadzu produces more lines of Ultra Performance Liquid Chromatography (defined in this way by Shimadzu Co.), in particular the 2 series of the instrument model called Nexera is an UPLC.

Line 161. The patient enrolment and Ethical committee grant have nothing to do with the title of that heading »chromatographic conditions«. On the other hand, some very important data IS missing: the column model, dimensions, the mobile phase composition at various time points or the gradient etc.

Thank you for your professional comment. Information on patient recruitment has been included in the new paragraph 2.7. As regards the further information concerning the analytical method, it is not considered necessary to include them, since the article has no methodological vocation, as well specified in the introduction.

Line 161. Which were the inclusion and exclusion criteria for the subjects?

Inclusion and exclusion criteria have been inserted in the new paragraph 2.7, thank you for the suggestion.

Line 172. Explain the sentence containing the phrase »..and derives was prevented«.

Thank you for raising this point of potential confusion, the sentence was reformulated.

Line 186. Please explain how was the flow combined from the chromatographic system with the flow of the infusion containing the reference standards. What is meant by the term »medium concentrations of phases«?

Thank you for your interest on our work. The text has been expanded (lines 191-194).

Line 197. The text refers to Table 2, which is missing in the manuscript.

We are sorry for the inconvenience. In the latest version, downloadable, the table 2 is visible to us. We will contact the editor to communicate the incompleteness of your version.

Line 199-202: Table 1 – please add accuracy data.

Thank you for the suggestion, the information has been added to Table 2.

Line 199-202, Table 1: How is LEV1 related to QC L?

The LEV1 was not corresponding to concentration of QC L. See Table 2. Thank you for your comment.

Line 199-202, Table 1 and Lines 193-195: In the cited literature and guidelines, there is no reference for standard deviation on repeatability sR, repeatability coefficient R, nor for Rsdr and RSDR.

Thank you for your professional comment. Those who practice validation of analytical methods use the repeatability limit as well as the deviation type of repeatability and that of reproducibility to evaluate the "goodness" of the data obtained in the respective analytical sessions. These parameters are contained in the relevant international technical standard (see for example ISO 21748:2017).

Line 199-202, Table 1: Which concentration level does the calculated recovery correspond to, QC-low, medium or high?

The concentration levels reported in Table 2 not corresponding directly to QC that were used in routine analysis. See the new Table 2 and thank you for your revision.

Line 208. Rephrase the sentence containing the unusual wording »gap minus«.

Thank you, the sentence has been reformulated.

Line 219. The Hubaux-Vos algorithm is not a part of ICH-M10 guidelines which the authors claim to have followed for their validation of the method.

The guidelines provide "rules" to be followed to optimize the different steps needed to validate a certain analytical method. So the guidelines say "what to do" does not say "how to do it". In the case of the LOD the same consideration applies. The authors for the evaluation and subsequent calculation decided to follow the Hubeaux-Vos algorithm.

Line 155: Please report the centrifugal force in correct terms (g-units, not RPM).

Thank you for the revision, the centrifugal force in correct terms (g-units) was reported at line 163.

Line 226, Figure 2: Please present also the chromatogram of internal standard and creatinine.

Thank you for the suggestion. The Figure 3 with the required chromatograms has been added.

Line 244: What are the levels for spiking (the data on QC-M is missing)?

The line 244 is the sequent <<... evaluated by r , where s represented sR or reproducibility conditions, . We are sorry, but what does that have to do with QC M?

Line 265: How was the method's robustness tested and where are the results?

Thank you for your revision. The data relating to robustness are not reported because the article does not have a methodological aim. For information the robustness has been evaluated as described in the homonymous paragraph following the ICH M10. The data was analyzed through the classic Yates matrix. In any case, this information is not essential for the handling of the subject matter of the manuscript.

Line 262-276: The cited ICH M10 guidelines do not propose the calculation of response factors for matrix effect evaluation, but the preparation of low and high QCs in blank matrix from 6 individuals. Therefore, the unspiked response should be subtracted from the spiked response, thereby obtaining the signal gained from spiking in each lot of matrix. In effect, this is similar to the response factor calculation, nevertheless it is better to strictly adhere to the guidelines if they are reported to be followed.

As mentioned above, the guidelines establish a criterion for performing a series of operations necessary to evaluate the different magnitudes involved in the analytical validation of a method. RF (response factor) is only a method of calculation that is necessary and complementary to what is required by these guidelines. Thank you for the time spent on our work.

Line 281: Consider the term sample stability instead of shelf life which is used for pharmaceutical and food industry products.

Thank you for the suggestion. The shelf-life terms was substituted with stability sample in each parts of manuscript.

Line 298: Where are the results from matrix effect study?

Thanl you for your revision, as explain above, the article does not have a methodological aim.

Line 324, table 3: what were the levels of creatinine in each group?

Thank you for your attention. Creatinine data are not the subject of this study. Creatinine concentration measurements are useful for the normalization of the concentration of orotic acid in the urinary matrix.

Line 330: figures 3 and 4, and table 3: Why were 30-50 and 50-70 groups combined in the male samples?

Thank you for yoor revision. The different distribution depends on the composition of homogeneous groups in terms of number of subjects and levels of oroticuria.

Lines 383-392: The authors should carefully re-evaluate what to include in their discussion regarding the differences in the absorption and metabolism in children in the ages from 0-10 years and the effects of menstrual cycle. Especially considering that male subjects 0-10 years had significantly higher levels of OA compared to other groups.

As mentioned above, this paper aims to give a starting point on the possible evaluations to understand the different expression of the biosynthesis of pyrimidines according to the different phases of life. This can be exemplified by the concentration levels of orotic acid in the urine. Therefore, it is necessary to perform other biochemical analyses.

Line 360: The inserted explanation of the Van Deemter model does not fit into the context of comparison between the presented method and other published methods, unless the authors conclude this is a crucial difference or benefit relative to other methods.

Thank you for your comment. Van Deemter’s model is the basis of the general theory of chromatographic separation. The parameters used such as the number and height of the theoretical plates are commonly used to verify chromatographic efficiency and monitor the analytical process over time, as already indicated in previous answers. This type of affixing makes it easy to assess the drift of the analytical process before it happens.
